# Rectus femoris electromyography signal clustering: Data-driven management of crouch gait in patients with cerebral palsy

**Mehrdad Davoudi**[1], **Firooz Salami**[1], **Robert Reisig**[1], **Dimitrios A. Patikas**[2], **Sebastian I. Wolf**[1] *

**1** Clinic for Orthopaedics, Heidelberg University Hospital, Heidelberg, Germany, **2** Laboratory of Neuromechanics, School of Physical Education and Sports Science at Serres, Aristotle University of Thessaloniki, Thessaloniki, Greece

* Sebastian.Wolf@med.uni-heidelberg.de

**Data Availability Statement:** All relevant data are within the manuscript and its Supporting information files.

## Abstract

This study aimed to investigate how electromyography (EMG) cluster analysis of the rectus femoris (RF) could help to better interpret gait analysis in patients with cerebral palsy (CP). The retrospective gait data of CP patients were categorized into two groups: initial examination (E1, 881 patients) and subsequent examination (E2, 377 patients). Envelope-formatted EMG data of RF were collected. Using PCA and a combined PSO-K-means algorithm, main clusters were identified. Patients were further classified into crouch, jump, recurvatum, stiff and mild gait for detailed analysis. The clusters (labels) were characterized by a significant peak EMG activity during mid-swing (L1), prolonged EMG activity during stance (L2), and a peak EMG activity during loading response (L3). Notably, L2 contained 76% and 92% of all crouch patients at E1 and E2, respectively. Comparing patients with a crouch gait pattern in L2-E1 and L2-E2, two subgroups emerged: patients with persistent crouch (G1) and patients showing improvement at E2 (G2). The minimum activity of RF during 20–45% of the gait was significantly higher ($p = 0.025$) in G1 than in G2. A greater chance of improvement from crouch gait might be associated with lower RF activity during the stance phase. Using our findings, we could potentially establish an approach to improve clinical decision-making regarding treatment of patients with CP.

## 1 Introduction

Cerebral palsy (CP) is a nonprogressive movement and posture disorder that develops in a fetus during pregnancy or infancy [1]. It is caused by an injury to the developing brain, which mostly happens before birth. Spastic CP is the most common form of the disease, which arises from damage to the motor cortex [1]. In this condition, muscles appear stiff and tight. Abnormal muscle tone and motor deficit affect the walking ability of patients with CP [2].

Surface electromyography (EMG) can be used to measure muscle activity in a noninvasive and clinically meaningful manner. In patients with CP, 3D gait analysis with simultaneous EMG measurements is often conducted to gain insight into muscle function as part of

**Funding:** German Research Foundation (DFG) (no: WO 1624/ 8-1). The Funders had no role in study design, data collection and analysis, decision to publish, or preparation of the manuscript."

prescribing treatment and evaluating treatment effect [3]. Combining EMG data of the rectus femoris (RF) muscle with kinematics, kinetics, and clinical data, Reinbold et al. demonstrated a method to predict the outcome of RF transfer surgery [4]. They concluded that a diminished range of knee flexion and a high activity of RF during swing phase, in addition to a positive RF spasticity (Ely) test, are the main factors for deciding to perform RF transfer surgery [4]. Patikas et al. [5] also suggested that EMG could be used to better interpret gait in children with hemiplegic spastic CP. They reported a prolonged activation of RF during the swing phase as a part of the underlying gait compensatory mechanisms in these patients. Additionally, they stated that EMG signals can describe the clinical condition of patients before and after surgery [6]. Particularly, several studies also reported a strong association between the EMG of RF and gait impairments in these patients [7–9]. EMG can also be utilized to develop improved control strategies for lower-limb exoskeletons, thereby enhancing motor function in patients with CP [10, 11].

Although the literature confirms the importance of EMG for treatment decision-making in patients with CP, there are still some limitations here, such as cross-talk, artifacts, and poor signal quality [12], for using these data in clinical settings. In addition, interpreting the results derived from these signals requires the expertise of a clinician and, currently, this is primarily conducted visually and qualitatively [13]. Consequently, there is a need for an approach that can assist clinicians by providing an objective analysis and interpretation of the EMG signals.

Some studies quantitively classified CP patients into different groups based on kinematics. Sutherland and Davids [14] classified common gait abnormalities of the knee in CP into four types: crouch, jump, recurvatum, and stiff knee. Rodda et al. [15] proposed an algorithm using a combination of 3D gait analysis, videos, and clinical examinations to classify the gait of patients with hemiplegia and diplegia. While these studies [16] have shown that kinematics-based grouping is helpful for treatment management and clinical decision-making in CP, a comprehensive EMG-based grouping system for these patients remains a challenge. The lack of access to an extensive EMG database definitively poses a significant problem for developing such systems.

Clustering analysis represents an analytical technique to group-unlabeled data for extracting meaningful information [17]. In recent studies, researchers used a K-means clustering algorithm as an unsupervised approach to classify the pathological gait patterns observed in patients with CP [18, 19]. Sangeux et al. [20] conducted a study using a large dataset of CP patients to compare sagittal gait patterns and K-means clustering. They introduced the "Plantar flexor–Knee extension (PFKE) couple index", which measures the distance of ankle and knee kinematics during 20 to 45% of the gait cycle relative to normative data. Their findings revealed a significant association between the traditional CP gait groups and the five clusters identified through the PFKE-based K-means algorithm. Additionally, they observed a correlation between the clusters and spasticity in the gastrocnemius-soleus muscles.

K-means is a powerful clustering algorithm that is widely used for various clustering problems. However, this method is associated with two significant limitations: 1) converging to a local minimum and 2) sensitivity to selecting the initial cluster centroids, which converges on the local rather than the global optima [21]. Therefore, the initial selection of cluster centroids plays a critical role in processing the K-means algorithm. This challenge can also be considered as the optimization of an "objective function" that effectively groups the points in the data space into clusters. To address this problem, researchers have proposed several methods that employ global optimization search algorithms to determine the initial points for the K-means algorithm [22]. Particle swarm optimization (PSO) is a good, nature-inspired and population-based, effective global optimization algorithm [23].

To the best of our knowledge, no clustering research has been conducted regarding EMG in patients with CP while walking. In the current study, we focused on EMG of the RF because of the importance of this muscle in the gait of these patients. Developing a hybrid PSO- K-means clustering algorithm, we aimed to analyze the EMG data of patients with CP and to evaluate the relationship between common CP gait abnormalities, changes in gait over time, and the identified clusters. Therefore, we hypothesize that the RF EMG patterns of patients with CP are not uniform and that they can be classified into clusters that might be linked to the clinical picture. Furthermore, examining these EMG patterns before and after treatment might help us identify EMG features that could aid in patient prognosis.

## 2 Materials and methods

### 2-1 Participants

The data analyzed in this study were part of a larger database established at the local University Clinics in the years 2000–2022 and derived from more than 2000 hemiplegia/diplegia patients with CP and about 350 typically developed individuals. This study was approved by the local ethics committee with the serial number S-243/2022. The retrospective gait data used for this study were primarily divided into two groups. The group of patients who visited the gait lab for the first time (E1, first examination) and those who visited a second time, too (E2, second examination). Therefore, it should be noted that the E2 patients in our study represent the same individuals examined at two different time points. The inclusion criteria were availability of EMG, kinematics and kinetics data for each subject, walking only barefoot without any assistive device, and classified as GMFCS level I, II, or III [16]. For hemiplegia patients, only the affected side was considered. After applying these criteria, 881 and 377 patients were recruited as E1 and E2, respectively, and 117 persons as typically developed (TD) (reference group). In addition, considering the hip, knee and ankle angle of the patients in sagittal plane during the gait, they were classified as crouch, jump, recurvatum, stiff knee, and mild gait [14, 24]. The characteristics of the participants are shown in Table 1.

### 2-2 Data acquisition

All subjects walked barefoot at a self-selected speed along a lane 15 m in length during data acquisition. Kinematics and kinetics were recorded using a twelve-camera 3D motion analysis system (VICON, Oxford Metrics Limited, UK) operating at 120 Hz and using three force plates (Kistler Instruments Co.), respectively. The skin-mounted markers were applied according to the protocol of Kadaba et al. [25] and the plug-in-gait model was chosen for analysis. Subsequently, gait parameters of at least seven strides were determined.

**Table 1. Demographic and descriptive data of the participants at their first (E1) and second (E2) examinations and also healthy individuals.**

| | E1 (n = 881) | E2 (n = 377) | TD (n = 117) |
|---|---|---|---|
| Age (years) | 16.8 ± 9.6 (58.3–3.3) | 16.9 ± 8.5 (54.2–3.7) | 21.7 ± 12.3 (46–6) |
| Height (cm) | 151.9 ± 37.1 (197–94) | 152.6 ± 18.2 (196–98) | 161.9 ± 19.8 (195–108) |
| Body mass (kg) | 46 ± 19.5 (125.1–13.6) | 46.8 ± 17.3 (101.8–14.5) | 66.4 ± 16.8 (91–19) |
| Sex (male/female) | 510/371 | 212/165 | 58/59 |
| CP type (diplegia/ hemiplegia) | 700/181 | 332/45 | |
| Subgroups (crouch/ jump/ recruvatum/ stiff/ mild) | 72/65/109/42/593 | 38/28/24/19/268 | |
| GMFCS level (I/ II/ III) | 140/130/15 | 59/67/3 | |
| Interval between examinations (years) | 2.4 ± 1.9 (13.5–0.1) | | |

The EMG data were recorded from eight lower extremity major muscles including the RF, right and left legs, using myon 320 (Myon AG, Schwarzenberg, CH). Bipolar surface adhesive electrodes (Blue Sensor, Ambu Inc., Glen Burnie, MD, USA) were placed on the targeted muscles, following the guidelines provided by SENIAM [26]. The distance between the electrodes was set at 2 cm [6]. To amplify the EMG signal, the Biovision EMG apparatus (Biovision Inc., Wehrheim, Germany) before 2013/2014 and via Delsys (Delsys Inc., Natick, MA, USA) after 2013/14, was utilized with a preamplification factor of ×5000.

Clinical examination was exclusively performed by two physiotherapists controlling each other. Knee extensor muscle strength was assessed according to the Medical Research Council (MRC) [27]. The spasticity of the RF was tested both by the Duncan-Ely test [28] and by the Tardieu test [29]. According to the MRC, muscle strength ranged from 5 (the strongest) to 1 (the weakest) The scale for spasticity ranged from 0 (no spasticity) to 4 (severe spasticity). More information about these (strength and spasticity) grading systems is available in our previous work [30].

## 2-3 Signal processing

The raw EMG data were then band-pass filtered (Butterworth filter with a cutoff frequency of 20–350 Hz), rectified, and the signal smoothed (Butterworth low-pass filter with a cutoff frequency of 9 Hz), amplitude-normalized to the mean of signal, time-normalized within one gait cycle (101 datapoint) in MatLab (The MathWorks, Inc. USA) [6]. The gait events were detected for both right and left sides for at least seven trials for each individual. The final EMG envelopes were calculated as the averaged time-normalized signals for all the valid strides. Further, six main features (mean, range, max, min, and their timing) from the EMG envelopes were extracted during ten gait phases [6]. These phases included the whole-gait-cycle, stance, swing, loading response, mid-stance, terminal stance, pre-swing, initial swing, mid-swing, and terminal swing phase. These features were used for further analysis.

To determine a magnitude that describes the deviation of a patient's EMG feature from the reference group, the norm-distance (NDi) was calculated according to [6] by Eq (1). ND was defined as the absolute difference between the i_th feature of the EMG of the patient p ($F_{pi}$) and the mean value of the same feature in the reference group ($\bar{F}_{ni}$), divided by the corresponding standard deviation within the reference group ($SD_{ni}$). This standardization process served as the initial input for subsequent steps and can potentially help as a data transformation method to identify meaningful clusters.

$$NDi = \frac{|F_{pi} - \bar{F}_{ni}|}{SD_{ni}} \tag{1}$$

## 2-4 Cluster analysis

We applied cluster analysis as an unsupervised stand-alone tool to gain insight into the data distribution, examine the distinct characteristics of each cluster, and prioritize specific clusters for subsequent analysis. Therefore, we applied two feature matrices with dimension of 881×60 (number of feature (6) × number of gait phases (10)) and 377×60 for E1 and E2 conditions, respectively. Prior to performing clustering analysis, we employed principal component analysis (PCA) to reduce the dimensionality of the input matrices [31]. The first principal components (PCs), which explained more than 96% of the total variance, were used for the clustering. The hybrid clustering algorithm was developed using MatLab software based on the details described in Supplementary material (S1 Appendix). In the initial stage, the PSO algorithm was employed for a global search to explore the possible optimal solutions to

predefine the number of clusters. The output of PSO served as the initial centroids for the K-means algorithm, which was then utilized to refine and generate the final result. Using the elbow method [32], the number of 'K' was determined. In this method, the changes in the sum of squared differences between the observations and their cluster (SSE) were analyzed by adding the number of clusters. The point at which there is a sharp change in the elbow curve indicates the K. In this study, we applied the elbow method on the E1 dataset to determine the K; then we set this for the E2 clustering as well.

## 2-5 Comparison between the clusters

The Pearson correlation (r) was applied to compare the averaged RF EMG of patients in the different clusters with that of the typically developed group. A more comprehensive investigation of the clusters (labels)' characteristics conducted through 1) extracting the average hip, knee, and ankle joint angles and moments of each cluster at E1 and E2 in the sagittal plane and 2) examining the population of the gait subgroups (crouch, jump, recurvatum, stiff knee, and mild) in each cluster. Descriptive statistics (mean and standard deviation) were used to compare the clinical examination data between the conditions.

## 2-6 Comparison between the groups

Considering the changes in the EMG cluster and gait kinematics of the patients from their second examination (E2), we set two cohorts of patients in E1 in an identified cluster (section 3) who were determined as crouch and who did not show any changes (G1) or showed improvement (G2) in their gait later according to E2. All of these patients underwent single-event multilevel surgery (SEMLS) between the examinations. Gait profile score (GPS) [33] was computed for the groups in both E1 and E2 to assess gait improvement.

To investigate the gait factors resulting in these different responses to the treatment, the EMG data of the RF muscle from G1 and G2 patients at E1 were compared by extracting six main features (section 2–3) during 20 to 45% of their gait cycle, as described by Sangeux et al. [20]. Applying the nonparametric Kruskal-Wallis test, we compared the features between G1 and G2 at E1 (p-Value = 0.05). Furthermore, we used statistical parametric mapping (SPM, www.spm1d.org) implemented in MatLab [34] to compare joint patterns. Fig 1 illustrates the procedure we used in this study in a flow chart.

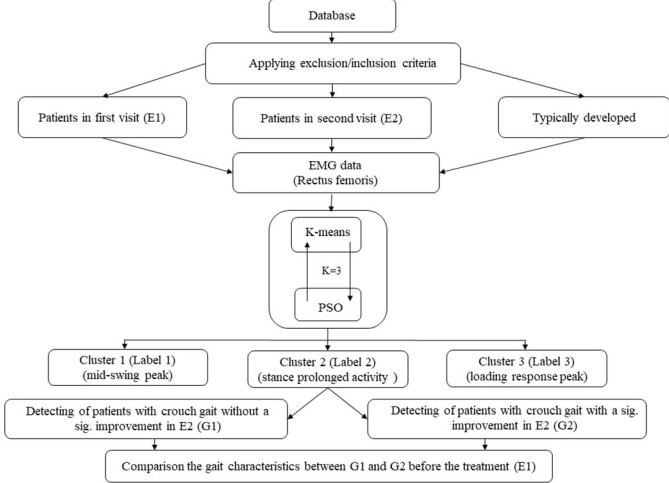

**Fig 1. Flowchart of the methodology used in this study.** EMG, electromyography; PSO, particle swarm optimization.

## 3 Results

Fig 2 shows an elbow plot with K = 3, representing the number of clusters (L1, L2, and L3). The first 25 PCs accounted for approximately 96.1% of the variance in the primary dataset and were utilized for the clustering analysis. Comparing the EMG of RF between the three identified clusters in two examinations (Fig 3A), they were characterized by a peak during mid-swing (L1), prolonged activity during (mid- and terminal) stance (L2), and a significant peak during loading response (L3). Notably, L2 contained 76% (55 out of 72) and 92% (35 out of 38) of all crouch patients at E1 and E2, respectively (Table 2). An excessive knee flexion during the stance phase of gait (Fig 3B), along with lack of an extension moment (Fig 3C), for patients detected as cluster 2, supported the sensitivity of our clustering to the crouch gait.

Correlation analysis (Table 2) revealed a similarity of more than 0.9 between the average EMG pattern of patients placed in L3 and the normal population, while L1 patients had the lowest coefficient by 39 and 53% at E1 and E2, respectively.

In addition, averaged clinical assessment of the patients (as presented in Table 2 descriptively) revealed that the L2 group had lower strength and higher levels of spasticity in their knee extensors than did the other two groups. Moreover, the number of patients with more severe motor impairment (GMFCS level III) was also higher in the L2 than in the L1 and L3 groups. Comparing G1 (16 patients, 32 limbs) and G2 (8 patients, 15 limbs), there was a significant improvement in the GPS of G2 at E2 (Table 3).

The SPM results did not show any systematic difference (p<0.05) between knee kinematics of G1 and G2 individuals at E1 (Fig 4A and 4B). Visually, however, the stance peak extension moment for both groups is at the same level (Fig 4C). As increased RF activity during stance was the main characteristic of cluster 2, a subjective comparison between the two groups in this phase (Fig 4A) showed a higher average EMG in G1-E1 than G2-E1. Statistically, as shown in Table 2, there is a systematic difference between the minimum activity of RF during 20–45% gait between the two groups (p = 0.025). Mean and maximum (features) were also lower for G2. We only reported the main three features with the lowest p-value.

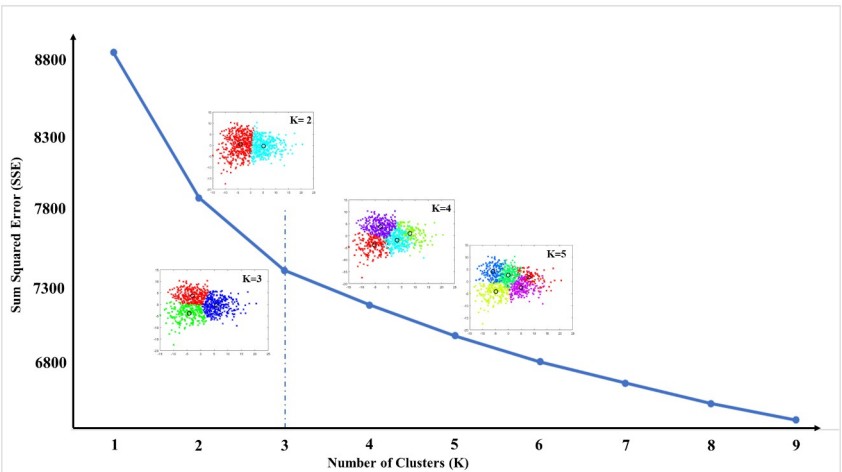

**Fig 2. Determination of the number of clusters using the elbow method.** The scatter plots show the distribution of input data (E1) in different clusters in which the axes are PC1 and PC2. Each color represents a cluster.

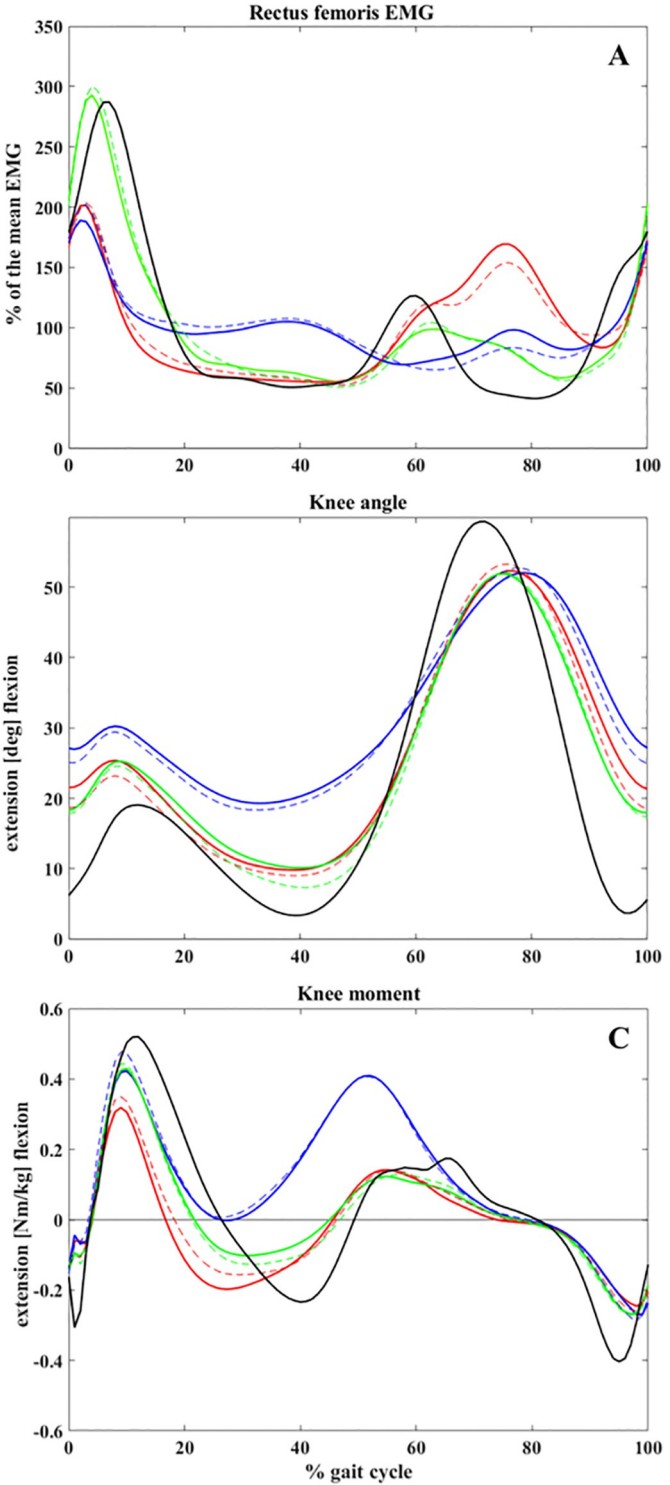

**Fig 3. Average rectus femoris electromyography (EMG) (A), knee kinematic (B) and knee kinetic (C) patterns for different clusters and examinations.** Red: L1, Blue: L2, Green: L3, Black: reference group, Solid lines: E1, Dashed lines: E2.

**Table 2. Number of patients with a gait abnormality, clinical examination data (mean ± SD), and number of patients with different GMFCS level in each cluster examination.**

| Conditions (Clusters-Examinations) | Gait subgroups count per condition | | | | | r-correlation to normal EMG | Clinical Examination Parameters | | | GMFCS level (I/ II/ III) |
|---|---|---|---|---|---|---|---|---|---|---|
| | Stiff | Recurvatum | Mild | Jump | Crouch | | Knee extensors strength | Rectus spasticity Tardieu | Rectus spasticity Duncan-Ely | |
| E1-L1 | 11 | 63 | 224 | 25 | 7 | 0.39 | 4.5 ± 0.6 | 0.66 ± 1 | 0.64 ± 0.5 | 57/40/2 |
| E1-L2 | 21 | 26 | 178 | 23 | 55 | 0.67 | 4.3 ± 0.6 | 1.1 ± 1 | 0.71 ± 0.4 | 19/45/11 |
| E1-L3 | 10 | 21 | 190 | 17 | 10 | 0.91 | 4.6 ± 0.5 | 0.44 ± 0.8 | 0.49 ± 0.5 | 64/45/2 |
| E2-L1 | 6 | 6 | 107 | 10 | 3 | 0.53 | 4.3 ± 0.6 | 0.68 ± 1 | 0.72 ± 0.5 | 30/20/0 |
| E2-L2 | 5 | 9 | 79 | 12 | 35 | 0.7 | 4.2 ± 0.6 | 0.96 ± 1 | 0.69 ± 0.5 | 10/29/2 |
| E2-L3 | 8 | 8 | 83 | 6 | 0 | 0.92 | 4.4 ± 0.6 | 0.44 ± 0.9 | 0.59 ± 0.5 | 19/18/1 |

**Table 3. Mean ± SD and statistical comparison for gait profile score (GPS) and rectus femoris electromyography (EMG) features at 20–45% gait between patients with crouch gait (cluster 2) measured before and after surgery (E1 and E2) and demonstrated no significant improvement (G1) or significant improvement during the second examination (G2).**

| | GPS | | | |
|---|---|---|---|---|
| | G1 | G2 | Normal | |
| E1 | 16.25 ± 2.6 | 18.4 ± 5.9 | 4.87 ± 1.09 | |
| E2 | 15.16 ± 3.9 | 14.3 ± 6.3 | | |
| p-value between E1 and E2 within each Group | 0.167 | **0.033*** | | |
| | Rectus EMG features in 20–45% gait cycle during E1 | | | |
| | G1 | G2 | p-value between G1 and G2 for each feature | Normal |
| Min 20–45% | 85.1 ± 20.2 | 69.2 ± 25.7 | **0.025*** | 38.3 ± 14.5 |
| Mean 20–45% | 110.8 ± 16.3 | 100.8 ± 23.4 | 0.144 | 56.7 ± 21 |
| Max 20–45% | 137.8 ± 19.1 | 128.1 ± 27.4 | 0.121 | 92.2 ± 36.8 |

*p-value<0.05

## 4 Discussion

Using an unsupervised hybrid PSO- K-means cluster analysis, three main groups were identified from EMG data of the RF in patients with CP. These clusters differed from each other in level of activity in swing (L1), stance (L2), and loading response (L3) (Fig 3A). Applying a pre-clustering standardization technique in combination with PCA, our clustering system could categorize the patients in relation to the deviation of their EMG results from those of a healthy population. Subsequently, a correlation of more than 90% was observed between one of the clusters (L3) and a normal EMG (Table 2). Descriptively, L3 patients showed a stronger knee extensor and a lower RF spasticity. On the other hand, patterns in patients identified as L1 correlated least with normal patterns (Table 2). It has been reported that in the majority of children with CP the RF is active during the mid-swing phase, when this muscle is normally inactive [35]. A high swing peak observed in the mean EMG of this cluster (L1) aligns with this typical feature of CP.

Furthermore, identifying a significant number of patients with a crouch gait as L2 (Table 2) is consistent with the prolonged activation of RF during stance phase, which is supported by the current literature [14]. In healthy individuals, quadriceps muscles are typically active for a small portion of the stance phase. However, in patients with crouch gait, the positioning of the ground reaction force behind the center of the knee joint requires the quadriceps to be engaged throughout the entire stance phase in order to maintain stability of the knee joint

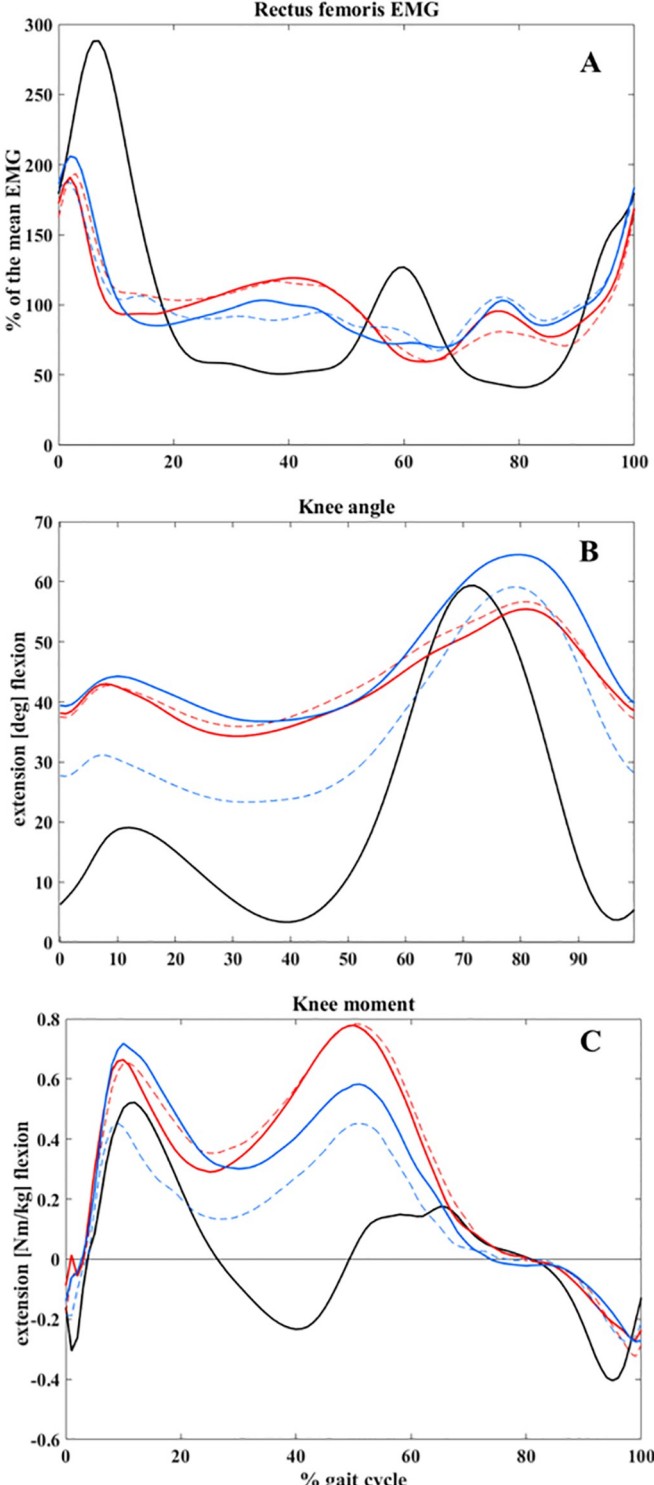

**Fig 4. Average rectus femoris electromyography (EMG) (A), knee kinematic (B) and knee kinetic (C) patterns for cluster 2 patients that did not improve (G1) or improved (G2) after surgery (E2) compared to their condition before surgery (E1).** Red: G1, Blue: G2, Solid line: E1, Dashed line: E2.

[36]. Table 1 demonstrates higher levels of RF and weaker knee extensor strength among individuals classified as L2 at E1 compared to L1 and L3. This cluster also exhibited a greater proportion of patients with a higher GMFCS level, which may support the notion that this EMG pattern might be linked to severity of the disability.

Investigating the relationship between EMG activity and altered kinematics in G1 patients with persistent crouch condition and G2 patients with significant GPS improvement, we observed (Table 3) that a lower minimum activity of RF during 20–45% of the gait cycle was a significant (p = 0.025) indicator of a better GPS at E2. However, the changes from E1 to E2 for sagittal knee kinematics of G1 and G2 were not significant. These findings suggest that EMG patterns are important for treatment decision-making, but that the effectiveness of kinematics, as a widely used clinical measure, is limited. The G1-G2 EMG comparison was made for the period 20–45% of the gait cycle. Sangeux et al. [20] developed an index to categorize CP subgroups by considering only this 25% of the gait cycle, mainly to avoid the loading response effects on the stance phase features. As the relationship between muscle EMGs and joint kinetics is nonlinear [37], it is difficult to find a direct explanation for the differences in the kinetics patterns of G1-G2 at E1 based on the RF activity.

Several algorithms are available in the literature to determine the optimal number of clusters [38]; however, in this study, the elbow curve at K = 3 was obvious and an exact biomechanical meaning for the clusters supports the algorithm results. For instance, each cluster showed a prominent activation in different gait phases: there was a cluster with a significant number of patients with crouch and a weaker RF, in addition to a cluster with patients with (correlated) normal EMG patterns. In this research, we used blind, unsupervised clustering without any prior information about the available EMG data. The only inclusion criteria applied were the availability of data from patients who walked barefoot and without any assistance. Applying such an algorithm on a large database (collected in our center) with more than 1000 examinations aided us in identifying general trends in the EMG data of CP patients. Furthermore, the term E2 in our study addressed the second examination of patients in whom a gait test (E1) had previously been performed in the lab. As a result, this does not apply to a before-after surgery scenario. However, it is important to note that these patients are typically engaged in an everyday training program. In the present study, the inclusion or exclusion criteria did not specifically consider the aspect of "treatment" initially. Crosstalk remains a major challenge in EMG studies, especially when attempting to isolate the activity of specific muscles. Particularly, the crosstalk from the surrounding vastus lateralis muscle on the activity of RF has recently been extensively reported as a common issue when employing surface EMG, especially in the presence of crouch gait observed in children with CP [39]. However, utilizing wire fine EMG, true activation of RF was noted in 30–45% of crouched gait cycles [39]. Our study specifically focuses on the 20–45% phase of gait in G1 and G2, addressing this unaffected segment of RF activity. Nevertheless, despite lacking access to an extensive database of wire fine EMG data from CP patients, we believe that surface EMG, as a non-invasive and clinically relevant measure of gait in individuals with CP, has the capacity to capture the primary characteristics of muscle activity in these patients.

To overcome the K-mean algorithm initialization problem, we developed a hybrid K-mean and PSO optimization approach. Raouafi et al. [40] employed discriminative analysis and K-means clustering techniques to develop a classification approach for determining upper limb disability levels in patients with CP. Their study utilized kinematic and EMG data from a cohort of 13 patients. Similarly, Hu et al. [41] introduced a mixed K-means and hierarchical clustering algorithm to categorize gait patterns in individuals with CP. While methodological studies on gait in CP exist, our study was the first attempt at clustering EMG data in CP using such an extensive clinical dataset. Therefore, the current research focused only on one muscle

(RF) and, further, the crouch condition to investigate the possibility of developing CP-EMG assessment approaches. Given that the vastus lateralis is particularly important during the stance phase and is often active when the rectus femoris is not, we suggest including this muscle in combination with the rectus femoris for further EMG analyses in patients with CP. Furthermore, the gait subgrouping algorithm we developed, which is based on the hip, knee, and ankle angles in the sagittal plane as outlined in references [14, 24], was limited to identifying patients with one of the following conditions: crouch, recurvatum, jump, stiff, or mild knee gait abnormalities. Although our algorithm was designed to detect the most severe condition as the primary gait issue for a patient, multiple gait abnormalities can occur simultaneously. For future research, we recommend analyzing EMG while considering the severity of each gait abnormality for individual patients.

We conclude that clustering EMG data of the rectus femoris has the potential to establish a novel threshold-based treatment decision-making approach for patients with CP, particularly those exhibiting crouch gait. From a clinical point of view, our study demonstrated that, as a rule of thumb for patients with crouch knee gait and minimal rectus femoris activity greater than 85% of the mean EMG during mid-stance of the gait cycle (as shown in Table 3), the crouch gait will not be corrected following the intervention. These findings seem promising and suggest that clustering analyses should be applied on datasets with more muscles. In addition, we mainly evaluated knee movement while other joints and gait abnormalities could also be included in future studies.

## Supporting information

**S1 Appendix.**
(DOCX)

**S1 Data.**
(XLSX)

## Acknowledgments

For the publication fee we acknowledge financial support by Heidelberg University.

## Author Contributions

**Conceptualization:** Firooz Salami, Sebastian I. Wolf.

**Data curation:** Firooz Salami, Robert Reisig.

**Formal analysis:** Mehrdad Davoudi, Firooz Salami.

**Funding acquisition:** Sebastian I. Wolf.

**Investigation:** Mehrdad Davoudi, Robert Reisig.

**Methodology:** Mehrdad Davoudi, Dimitrios A. Patikas.

**Software:** Mehrdad Davoudi, Firooz Salami.

**Supervision:** Sebastian I. Wolf.

**Validation:** Firooz Salami, Robert Reisig, Dimitrios A. Patikas.

**Visualization:** Mehrdad Davoudi.

**Writing – original draft:** Mehrdad Davoudi.

**Writing – review & editing:** Firooz Salami, Robert Reisig, Dimitrios A. Patikas, Sebastian I. Wolf.

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
