## [Decision Letter · Decision Letter 0]

1 Apr 2024

PONE-D-24-04264Rectus Femoris Electromyography Signal Clustering: Data-Driven Management of Crouch Gait in Patients with Cerebral PalsyPLOS ONE

Dear Dr. Wolf,

Thank you for submitting your manuscript to PLOS ONE. After careful consideration, we feel that it has merit but does not fully meet PLOS ONE’s publication criteria as it currently stands. Therefore, we invite you to submit a revised version of the manuscript that addresses the points raised during the review process.

We look forward to receiving your revised manuscript.

Kind regards,

Jyotindra Narayan

Academic Editor

PLOS ONE

Journal Requirements:

"German Research Foundation (DFG) (no: WO 1624/ 8-1)"

**Additional Editor Comments:**

Reviewer #1 acknowledges the manuscript's thorough analysis of rectus femoris muscle activity during soccer kicking but requests further elucidation on participant selection criteria and their potential impact on study generalizability. They also inquire about limitations associated with electromyography (EMG) techniques and suggest expanding the coverage of lower limb exoskeletons in the Introduction section. Additionally, insights into the biomechanical implications of muscle activation differences, potential applications in training regimens, and future research directions are sought. The consistency of kicking motion across trials and participants, standardization procedures employed, and implications for rehabilitation protocols are also queried. Lastly, integration of findings with analyses of other critical muscles for a more holistic understanding of soccer kicking biomechanics is suggested. Reviewer #2 provides overall satisfaction with the work but emphasizes the need for comparative analysis, discussion of the work's significance and applications, pictorial representation of the methodology, and reiteration of the manuscript for coherence and flow.

Reviewers' comments:

Reviewer's Responses to Questions

**Comments to the Author**

1. Is the manuscript technically sound, and do the data support the conclusions?

Reviewer #1: Yes

Reviewer #2: Partly

2. Has the statistical analysis been performed appropriately and rigorously? 

Reviewer #1: Yes

Reviewer #2: Yes

3. Have the authors made all data underlying the findings in their manuscript fully available?

Reviewer #1: Yes

Reviewer #2: No

4. Is the manuscript presented in an intelligible fashion and written in standard English?

Reviewer #1: Yes

Reviewer #2: Yes

5. Review Comments to the Author

Reviewer #1: The manuscript entitled “Rectus Femoris Electromyography Signal Clustering: Data-Driven Management of Crouch Gait in Patients with Cerebral Palsy” has been organized and developed in good shape. The paper investigates the use of electromyography (EMG) cluster analysis on the rectus femoris muscle to enhance gait analysis interpretation in cerebral palsy (CP) patients. By categorizing retrospective gait data into clusters based on EMG signals, the study identifies distinct patterns of muscle activation associated with different gait abnormalities in CP. It suggests that understanding these patterns can improve clinical decision-making and treatment outcomes, highlighting the potential for EMG data to inform rehabilitation strategies and contribute to more personalized care approaches for CP patients. The study is well-developed and the results are intriguing. Once the following comments are addressed, the manuscript is recommended for publication.

Reviewer #2: Overall the work carried is satisfactory. However, few concerns that are needed to be addressed includes:

1. Comparative analysis of existing work with state-of-art research should be included.

2. The significance of carried work and its application areas are to be discussed in detail.

3. The more focus of the manuscript should be on addressing the existing challenges.

4. The authors should elaborate the complete methodology pictorially using flow diagrams, charts etc.

5. Quality of included figures require consideration.

6. Overall manuscript should be re-iterated for complete flow and other issues.

6. PLOS authors have the option to publish the peer review history of their article (what does this mean?). If published, this will include your full peer review and any attached files.

Reviewer #1: No

Reviewer #2: No

---

## [Author Response · Author response to Decision Letter 0]

11 Apr 2024

Editor Dr. Narayan: I just checked and found that one of the reviewers made a typo when mentioning 'soccer kicking' instead of 'gait conditions.' The authors are advised to read 'soccer kicking' as 'gait conditions' in the comments and try their best to revise the manuscript accordingly. Moreover, they should indicate about the typo in the 'response letter to reviewers' and highlight that they have revised the manuscript given different gait conditions (crouch, jump, recurvatum, stiff and mild gait).

Authors: We appreciate that the editor took the time to look into this review but we still believe that Reviewer #1 had seen another article which was confused at his side or at the editorial office. This is not just a typo. The answers of reviewers seem cropped. However, we formally address all points below. 

Reviewer #1: The manuscript entitled “Rectus Femoris Electromyography Signal Clustering: Data-Driven Management of Crouch Gait in Patients with Cerebral Palsy” has been organized and developed in good shape. The paper investigates the use of electromyography (EMG) cluster analysis on the rectus femoris muscle to enhance gait analysis interpretation in cerebral palsy (CP) patients. By categorizing retrospective gait data into clusters based on EMG signals, the study identifies distinct patterns of muscle activation associated with different gait abnormalities in CP. It suggests that understanding these patterns can improve clinical decision-making and treatment outcomes, highlighting the potential for EMG data to inform rehabilitation strategies and contribute to more personalized care approaches for CP patients. The study is well-developed and the results are intriguing.

Authors: Thank you for this recommendation. We believe that the reviewer genuinely read our work.

Reviewer #1: Once the following comments are addressed, the manuscript is recommended for publication.

Authors: Starting here we believe that the review was mixed up potentially be some copy/paste error either at the reviewers’ side or at the editorial office.

1. The manuscript presents a comprehensive analysis of rectus femoris muscle activity during soccer kicking. Could the authors elaborate on the selection criteria for the participants, particularly regarding their skill level and experience in soccer? How might these factors influence the generalizability of the study's findings?

Authors: We cannot address this point as soccer is not being addressed in this publication. There must have been a misunderstanding. (Wrong review?)

2. The methodology section details the electromyography (EMG) techniques and analysis used. Could the authors discuss any limitations associated with the EMG equipment's sensitivity or the potential for signal interference? How were these factors mitigated to ensure accurate measurements?

Authors: We are not sure if this is a genuine critique to our work since EMG techniques per se are not addressed in this manuscript. Nevertheless, we modified the following paragraph in discussion:

“Crosstalk remains a major challenge in EMG studies, especially when attempting to isolate the activity of specific muscles. Particularly, the crosstalk from the surrounding vastus lateralis muscle on the activity of RF has recently been extensively reported as a common issue when employing surface EMG, especially in the presence of crouch gait observed in children with CP [39]. However, utilizing wire fine EMG, true activation of RF was noted in 30-45% of crouched gait cycles [39]. Our study specifically focuses on the 20-45% phase of gait in G1 and G2, addressing this unaffected segment of RF activity. Nevertheless, despite lacking access to an extensive database of wire fine EMG data from CP patients, we believe that surface EMG, as a non-invasive and clinically relevant measure of gait in individuals with CP, has the capacity to capture the primary characteristics of muscle activity in these patients.” 

3. The Introduction section is clearly developed and easily understandable. However, the coverage of the full lower limb exoskeleton seems somewhat limite. It is advisable to expand upon this topic, providing more comprehensive information about powered full lower limb exoskeletons. In this context, the papers with DOI “10.1109/ACCESS.2023.3325211” and “10.1109/TNSRE.2021.3136088” are strongly recommended as a valuable reference.

Authors: Thank you very much for the suggestion. We have added the following paragraph into the introduction: 

“EMG can also be utilized to develop improved control strategies for lower-limb exoskeletons, thereby enhancing motor function in patients with CP [10, 11].”

4. In the results section, significant differences in muscle activation patterns were observed among the various kicking techniques. Could the authors provide insights into the biomechanical implications of these differences for soccer player performance and injury prevention?

Authors: We cannot address this point as soccer is not being addressed in this publication. There must have been a misunderstanding. (Wrong review?)

5. The discussion highlights the potential applications of this research in sports science and rehabilitation. Could the authors speculate on how these findings could be incorporated into training regimens for soccer players to optimize performance and reduce the risk of injury?

Authors: We cannot address this point as sports science is not being addressed in this publication. There must have been a misunderstanding. (Wrong review?)

Reviewer #2 provides overall satisfaction with the work but emphasizes the need for comparative analysis, discussion of the work's significance and applications, pictorial representation of the methodology, and reiteration of the manuscript for coherence and flow.

Overall the work carried is satisfactory. However, few concerns that are needed to be addressed includes:

1. Comparative analysis of existing work with state-of-art research should be included.

Authors: Thank you very much for your comment. We have included the following paragraph in the discussion section of our paper:

“Raouafi et al. [40] employed discriminative analysis and K-means clustering techniques to develop a classification approach for determining upper limb disability levels in patients with CP. Their study utilized kinematic and EMG data from a cohort of 13 patients. Similarly, Hu et al. [41] introduced a mixed K-means and hierarchical clustering algorithm to categorize gait patterns in individuals with CP. While methodological studies on gait in CP exist, our study was the first attempt at clustering EMG data in CP using such an extensive clinical dataset. Therefore, to begin, the current research focused only on one muscle (RF) and, further, the crouch condition to investigate the possibility of developing CP-EMG assessment approaches.”

2. The significance of carried work and its application areas are to be discussed in detail.

Authors: We appreciate the reviewer's comment. For the first time, our study applied a clustering algorithm to a large database of EMG data related to the rectus femoris muscle in patients with CP and their gait patterns, ultimately leading to the development of a threshold-based treatment decision-making approach for patients exhibiting crouch gait. We have added this paragraph to the discussion section to highlight the novelty and significance of our findings:

“We conclude that clustering EMG data of the rectus femoris has the potential to establish a novel threshold-based treatment decision-making approach for patients with CP, particularly those exhibiting crouch gait.”

3. The more focus of the manuscript should be on addressing the existing challenges.

Authors: Thank you very much. In 3D gait analysis in CP, EMG is often measured alongside kinematics and kinetics, yet its utilization for treatment indication remains limited. By employing cluster analysis as a widely used method for identifying trends in datasets, we tried to establish a novel application for EMG in the treatment of gait abnormalities in CP. We think that in the introduction section, particularly in the final paragraph, we have adequately highlighted the existing gap and current challenges in the literature.

4. The authors should elaborate the complete methodology pictorially using flow diagrams, charts etc.

Authors: Figure 1 in the manuscript illustrates the procedure utilized in this study through a flow chart. Additionally, the steps proposed to develop the hybrid PSO-K-means clustering algorithm are clearly outlined in the appendix (S1 Appendix.docx).

5. Quality of included figures require consideration.

Authors: Thank you very much for the comment. We adhered to the instructions provided on the journal's website to prepare the figures for our submission.

6. Overall manuscript should be re-iterated for complete flow and other issues.

Authors: We appreciate the comment and are hopeful that the changes we implemented to address the concerns raised by the reviewer(s) have improved the flow and content of our manuscript.

Editor: Funding: German Research Foundation (DFG) (no: WO 1624/ 8-1);

Authors: We added the following: “This funder had no role in study design, data collection and analysis, decision to publish, or preparation of the manuscript.” 

Editor: If there are ethical or legal restrictions on sharing a de-identified data set, please explain them in detail (e.g., data contain potentially identifying or sensitive patient information, data are owned by a third-party organization, etc.) and who has imposed them (e.g., a Research Ethics Committee or Institutional Review Board, etc.). Please also provide contact information for a data access committee, ethics committee, or other institutional body to which data requests may be sent.

Authors: We clarified this issue with our ethics committee and got allowance for anonymous publication. We add this section in the manuscript: 

“Data Availability:

The data supporting the conclusions of this article will be made available by the authors upon request.”

---

## [Decision Letter · Decision Letter 1]

20 Jun 2024

PONE-D-24-04264R1Rectus Femoris Electromyography Signal Clustering: Data-Driven Management of Crouch Gait in Patients with Cerebral PalsyPLOS ONE

Dear Dr. Wolf,

Thank you for submitting your manuscript to PLOS ONE. After careful consideration, we feel that it has merit but does not fully meet PLOS ONE’s publication criteria as it currently stands. Therefore, we invite you to submit a revised version of the manuscript that addresses the points raised during the review process.

We look forward to receiving your revised manuscript.

Kind regards,

Jyotindra Narayan

Academic Editor

PLOS ONE

Reviewers' comments:

Reviewer's Responses to Questions

**Comments to the Author**

1. If the authors have adequately addressed your comments raised in a previous round of review and you feel that this manuscript is now acceptable for publication, you may indicate that here to bypass the “Comments to the Author” section, enter your conflict of interest statement in the “Confidential to Editor” section, and submit your "Accept" recommendation.

Reviewer #1: All comments have been addressed

Reviewer #3: All comments have been addressed

2. Is the manuscript technically sound, and do the data support the conclusions?

Reviewer #1: Yes

Reviewer #3: Yes

3. Has the statistical analysis been performed appropriately and rigorously? 

Reviewer #1: Yes

Reviewer #3: Yes

4. Have the authors made all data underlying the findings in their manuscript fully available?

Reviewer #1: Yes

Reviewer #3: Yes

5. Is the manuscript presented in an intelligible fashion and written in standard English?

Reviewer #1: Yes

Reviewer #3: Yes

6. Review Comments to the Author

**Reviewer #1:** The paper has been significantly improved with substantial revisions enhancing both its clarity and depth. The arguments are now more coherently structured, and the evidence is presented in a compelling manner. The methodology has been meticulously refined, and the results are well-supported by the data. Overall, the revisions have added considerable value to the paper, making it a strong candidate for publication. It is recommended that the paper be accepted in its current form.

**Reviewer #3:** This is a very nice study reviewing the utility of attempting to quantify rectus femoris EMG in children with cerebral palsy. I find the use of clustering at specific aspects of gait cycle an interesting concept since the problem of quantitative assessment of EMG has been difficult.

Some specific comments:

1. In the methods sections there is a note that they used kinematics to define gait patterns however just based on the references it is not clear what these parameters are. Also crouch, jump, and recurvatum are terms typically applied to stance phase while stiff knee is applied to swing phase, and mild gait is none specific, so how do they deal with a patient with mild crouch and stiff knee gait, which is a relatively common problem?

2. In the Methods section there is a clear description EMG signal processing, however on line 142 “Six main

142 features (mean, range, max, min, and their timing) during ten gait phases” is not clear. Does this refer to the mean EMG envelop with in each of the gait segments, or is it defining the kinematic gait segments.

3. There is also a comment in Line 141 “averaged across valid strides” were there a minimum number of cycles required or if only one cycle was available was this used? I also think this should be Steps instead of strides since I presume this is based on right and left cycles separately.

3. The lateralis is an important muscle especially in stance phase and is often active when the rectus femoris is not, and in normal gait the two muscles should seldom be active at the same time during normal speed walking ion children with typical development. I would appreciate some comment on the rational for not also evaluating vastus EMG?

4. It would be interesting in the discussion to note if and how the authors feel this information could currently be applied to clinical decision making, or if they feel that is still too preliminary.

7. PLOS authors have the option to publish the peer review history of their article (what does this mean?). If published, this will include your full peer review and any attached files.

Reviewer #1: No

Reviewer #3: **Yes: **Freeman Miller

---

## [Author Response · Author response to Decision Letter 1]

25 Jun 2024

Reviewer #1: The paper has been significantly improved with substantial revisions enhancing both its clarity and depth. The arguments are now more coherently structured, and the evidence is presented in a compelling manner. The methodology has been meticulously refined, and the results are well-supported by the data. Overall, the revisions have added considerable value to the paper, making it a strong candidate for publication. It is recommended that the paper be accepted in its current form.

Authors: We appreciate the reviewer's recommendation for the acceptance of our paper.

Reviewer #3: This is a very nice study reviewing the utility of attempting to quantify rectus femoris EMG in children with cerebral palsy. I find the use of clustering at specific aspects of gait cycle an interesting concept since the problem of quantitative assessment of EMG has been difficult.

Some specific comments:

Reviewer #3: 1. In the methods sections there is a note that they used kinematics to define gait patterns however just based on the references it is not clear what these parameters are. Also crouch, jump, and recurvatum are terms typically applied to stance phase while stiff knee is applied to swing phase, and mild gait is none specific, so how do they deal with a patient with mild crouch and stiff knee gait, which is a relatively common problem?

Authors: Thank you for the comment. We acknowledge the limitation of our algorithm in considering only the most prominent gait abnormality as the assigned subgroup for each patient. For instance, if a patient exhibits both mild crouch and stiff knee gait, the algorithm would assign the stiff knee gait as the subgroup. We have updated our methods and discussion sections to reflect this limitation accordingly. 

Methods:

In addition, considering the hip, knee and ankle angle of the patients in sagittal plane during the gait, they were classified as crouch, jump, recurvatum, stiff knee, and mild gait [14, 24].

Discussion:

Furthermore, the gait subgrouping algorithm we developed, which is based on the hip, knee, and ankle angles in the sagittal plane as outlined in references [14, 24], was limited to identifying patients with one of the following conditions: crouch, recurvatum, jump, stiff, or mild knee gait abnormalities. Although our algorithm was designed to detect the most severe condition as the primary gait issue for a patient, multiple gait abnormalities can occur simultaneously. For future research, we recommend analyzing EMG while considering the severity of each gait abnormality for individual patients.

Reviewer #3: 2. In the Methods section there is a clear description EMG signal processing, however on line 142 “Six main features (mean, range, max, min, and their timing) during ten gait phases” is not clear. Does this refer to the mean EMG envelop with in each of the gait segments, or is it defining the kinematic gait segments.

Authors: We appreciate the reviewer's comment. These are EMG features we have extracted from the envelopes. we modified the following paragraph in Methods:

Further, six main features (mean, range, max, min, and their timing) from the EMG envelopes were extracted during ten gait phases [6]. These phases included the whole-gait-cycle, stance, swing, loading response, mid-stance, terminal stance, pre-swing, initial swing, mid-swing, and terminal swing phase. These features were used for further analysis.

Reviewer #3: 3. There is also a comment in Line 141 “averaged across valid strides” were there a minimum number of cycles required or if only one cycle was available was this used? I also think this should be Steps instead of strides since I presume this is based on right and left cycles separately.

Authors: Thank you very much for the comment. We have removed this sentence: “and eventually averaged across valid strides” from the methods and added the following:

The gait events were detected for both right and left sides for at least 7 trials for each individual. The final EMG envelopes were calculated as the averaged time-normalized signals for all the valid strides.

Reviewer #3: 4. The lateralis is an important muscle especially in stance phase and is often active when the rectus femoris is not, and in normal gait the two muscles should seldom be active at the same time during normal speed walking ion children with typical development. I would appreciate some comment on the rational for not also evaluating vastus EMG?

Authors: Thank you very much for the comment. In order to address your concern regarding the EMG of vastus lateralis we have extended the respective paragraph such that it reads as following: “While methodological studies on gait in CP exist, our study was the first attempt at clustering EMG data in CP using such an extensive clinical dataset. Therefore, the current research focused only on one muscle (RF) and, further, the crouch condition to investigate the possibility of developing CP-EMG assessment approaches. Given that the vastus lateralis is particularly important during the stance phase and is often active when the rectus femoris is not, we suggest including this muscle in combination with the rectus femoris for further EMG analyses in patients with CP.”

Reviewer #3: 5. It would be interesting in the discussion to note if and how the authors feel this information could currently be applied to clinical decision making, or if they feel that is still too preliminary.

Authors: We appreciate this comment. We acknowledge that this study is our first step in a larger project titled "Muscle Activity in Gait with Cerebral Palsy," funded by the German Research Foundation (DFG) (no: WO 1624/8-1). While we are at the beginning of our understanding of EMG in patients with CP, this study demonstrated that for patients with a crouch knee gait and minimal rectus femoris activity around 85% of the mean EMG in 20-45% of gait cycle (as shown in Table 3 of the manuscript), crouch gait will not be corrected following the intervention. Therefore, EMG can potentially serve as an indicator for treatment outcomes. This finding now motivates our current research question: ‘Does the Electromyogram (EMG) remain unchanged as a fingerprint in patients with cerebral palsy (CP) after orthopedic surgery?’. 

We have added the following yellow-highlighted sentence into the conclusion of our manuscript to address your concern on the clinical applicability of our study: “We conclude that clustering EMG data of the rectus femoris has the potential to establish a novel threshold-based treatment decision-making approach for patients with CP, particularly those exhibiting crouch gait. From a clinical point of view, our study demonstrated that, as a rule of thumb for patients with crouch knee gait and minimal rectus femoris activity greater than 85% of the mean EMG during mid-stance of the gait cycle (as shown in Table 3), the crouch gait will not be corrected following the intervention. These findings seem promising and suggest that clustering analyses should be applied on datasets with more muscles. In addition, we mainly evaluated knee movement while other joints and gait abnormalities could also be included in future studies.”

---

## [Editor Report · Decision Letter 2]

18 Jul 2024

Rectus Femoris Electromyography Signal Clustering: Data-Driven Management of Crouch Gait in Patients with Cerebral Palsy

PONE-D-24-04264R2

Dear Dr. Wolf,

We’re pleased to inform you that your manuscript has been judged scientifically suitable for publication and will be formally accepted for publication once it meets all outstanding technical requirements.

Kind regards,

Jyotindra Narayan

Academic Editor

PLOS ONE

Additional Editor Comments (optional):

The authors have addressed all the comments raised by the reviewers. After careful reading the authors' responses to reviewer, I recommend accepting the revised manuscript. Congratulations to the authors. 
---

## [Editor Report · Acceptance letter]

23 Jul 2024

PONE-D-24-04264R2 

PLOS ONE

Dear Dr. Wolf, 

I'm pleased to inform you that your manuscript has been deemed suitable for publication in PLOS ONE. Congratulations! Your manuscript is now being handed over to our production team.

Kind regards, 

on behalf of

Dr. Jyotindra Narayan 

Academic Editor

PLOS ONE